# Clinical, Histopathologic, and Immunohistochemical Features of Patients with IgG/IgA Pemphigus

**DOI:** 10.3390/biomedicines10051197

**Published:** 2022-05-22

**Authors:** Yung-Tsu Cho, Ko-Ting Fu, Kai-Lung Chen, Yih-Leong Chang, Chia-Yu Chu

**Affiliations:** 1Department of Dermatology, National Taiwan University Hospital and National Taiwan University College of Medicine, Taipei 100, Taiwan; yungtsucho@gmail.com (Y.-T.C.); sunnine2008@gmail.com (K.-T.F.); klnchen1017@gmail.com (K.-L.C.); 2Graduate Institute of Pathology, National Taiwan University College of Medicine, Taipei 100, Taiwan; 3Department of Dermatology, Fu Jen Catholic University Hospital, Fu Jen Catholic University, New Taipei City 243, Taiwan; 4Department of Pathology, National Taiwan University Cancer Center, Taipei 106, Taiwan

**Keywords:** autoimmune bullous disease, pemphigus, IgG/IgA pemphigus, IL-8, MMP-9, immunohistochemistry

## Abstract

Pemphigus is an autoantibody-mediated blistering disease. In addition to conventional pemphigus vulgaris and pemphigus foliaceus, several other types have been reported. Among them, IgG/IgA pemphigus is less well defined and seldom reported. To characterize the clinical, histopathologic, and immunohistochemical presentation of IgG/IgA pemphigus, we retrospectively identified 22 patients with the disease at a referral center in Taiwan. These patients showed two types of skin lesion: annular or arciform erythemas with blisters or erosions (45.5%) and discrete erosions or blisters such as those in conventional pemphigus (54.5%). Mucosal involvement was found in 40.9%. Histopathologic analysis identified acantholysis (77.3%) and intra-epidermal aggregates of neutrophils (40.9%) and eosinophils (31.8%). Direct immunofluorescence studies showed IgG/IgA (100%) and C3 (81.8%) depositions in the intercellular space of the epidermis. In immunohistochemical staining, patients with IgG/IgA pemphigus demonstrated significantly higher levels of epidermal expression of interleukin-8 and matrix metalloproteinase-9 than those with conventional pemphigus (*p* < 0.05). In conclusion, although IgG/IgA pemphigus is heterogeneous in presentation, it shows characteristic features that are different from other forms of pemphigus and should be considered a distinct type of pemphigus.

## 1. Introduction

Pemphigus is an autoantibody-mediated autoimmune blistering disease that can be life-threatening, with a one-year mortality rate of 4.2–8% [1,2]. The disease is mainly caused by anti-desmoglein (Dsg)-3 and/or anti-Dsg1 antibodies, which cause acantholysis in the epithelium or epidermis with subsequent blisters or erosions. The main types of pemphigus are pemphigus vulgaris (PV) and pemphigus foliaceus (PF), diagnosed based on the involvement of mucosal areas and the depth of skin erosions [3]. In addition to these conventional types of pemphigus, there are several less common subtypes, including pemphigus herpetiformis, pemphigus vegetans, IgA pemphigus, paraneoplastic pemphigus, and the even more rare subtype, IgG/IgA pemphigus [4].

IgG/IgA pemphigus is seldom reported in the literature, and there have only been a few case series to date [5,6,7]. The most accepted definition of IgG/IgA pemphigus is that it shows the presence of cell surface IgG and IgA on direct immunofluorescence (DIF) and/or evidence of circulating cell surface IgG and IgA on indirect immunofluorescence (IIF) or enzyme-linked immunosorbent assay (ELISA) [5,6,7]. However, the characteristics of IgG/IgA pemphigus continue to be debated. Whether IgG/IgA pemphigus is a special form of conventional PV and PF, a transitional type between IgG pemphigus and IgA pemphigus, or a distinct disorder of heterogeneous presentation remains to be determined [5,6,7]. In addition, studies examining the immunological features of skin lesions in patients with IgG/IgA pemphigus are still lacking.

Therefore, to better characterize patients with IgG/IgA pemphigus, we retrospectively collected and reviewed data for patients with IgG/IgA pemphigus. We also performed immunohistochemical (IHC) staining of potential biomarkers to compare expression between patients with IgG/IgA pemphigus and those with conventional pemphigus in order to identify the characteristics of the disease.

## 2. Materials and Methods

### 2.1. Patient Selection

We retrospectively selected eligible cases treated at a tertiary referral center in Northern Taiwan from 1999 to 2014. The study was approved by the hospital ethics committee (IRB number: 201811054RIND). Patients with IgG/IgA pemphigus were defined as those showing the presence of cell surface IgG and IgA on DIF and/or IIF [5]. Data on demographics, clinical presentation, and disease severity defined using the pemphigus disease area index (PDAI) [8], histopathologic features, results of DIF/IIF, and laboratory findings were collected for eligible cases. Skin specimens of patients with conventional pemphigus, i.e., those with typical presentation of PV or PF caused by only IgG autoantibodies, whose disease severity was similar to that of patients with IgG/IgA pemphigus during the same period were collected for comparison of IHC stains.

### 2.2. Immunohistochemical Staining

Formalin-fixed paraffin-embedded blocks of tissue from patients with IgG/IgA pemphigus or conventional pemphigus were cut into 5 μm thick sections for IHC staining. These sections were stained with myeloperoxidase (MPO) (ab9535; Abcam, Cambridge, UK), C5a (ab11877; Abcam, Cambridge, UK), CD89 (ab124717; Abcam, Cambridge, UK), interleukin (IL)-8 (ab18672; Abcam, Cambridge, UK), IL-17 (ab79056; Abcam, Cambridge, UK), and matrix metalloproteinase (MMP)-9 (ab76003; Abcam, Cambridge, UK). After autoclaving for antigen retrieval, staining was performed according to the manufactures’ instructions. We used secondary antibodies conjugated to peroxidase with 3-amino-9-ethylcarbazole to detect the expression of these molecules. The sections were randomly examined and evaluated blind by a dermatologist (YTC) and a pathologist (YLC). The expression of these molecules in the epidermis was categorized from grade 0 to grade 3 based on staining intensity (Appendix A). The number of positive infiltrating cells in the dermis was calculated by averaging the values of three randomly selected high-power fields for each specimen.

### 2.3. Statistical Analysis

Descriptive statistics were obtained for the demographic data. Chi-square tests were used to compare clinical presentation and histological features between the two different types of skin lesions in the patients with IgG/IgA pemphigus. For the comparisons of IHC stains between patients with IgG/IgA pemphigus and those with conventional pemphigus, categorical data were analyzed using chi-square tests, and continuous measurements were analyzed using the Mann–Whitney *U* test. All analyses were performed using Excel 2007 (Microsoft Corp, Redmond, WA, USA). A *p*-value less than 0.05 was considered statistically significant.

## 3. Results

### 3.1. Demographics and Clinical Presentations of Patients with IgG/IgA Pemphigus

Twenty-two patients with IgG/IgA pemphigus, including 10 males and 12 females, were identified and recruited (Table 1). The median age was 48.5 years (range 23–82 years). Most patients showed moderate to severe disease based on a median PDAI of 19 (range 6–47). IIF showed positive IgG depositions in 16 cases (72.7%), with a median titer of 1:20. In the case of comorbidities, four patients (18.2%) had hypertension, two patients (9.1%) had coronary artery disease, two patients (9.1%) had chronic urticaria, and two patients (9.1%) had a malignancy.

Mucosal lesions were mainly oral ulcers and were found in nine patients (40.9%). Cutaneous lesions were unique and could be categorized into two main types of presentation. One type of presentation was annular or arciform lesions, and it is found in 10 patients (45.5%) (Figure 1a,b,d,e). These patients presented with annular or arciform, erythematous, crusted, or eroded patches and edematous plaques. Some of these patches or plaques had vesicles or occasional pustules at the margins. The other type of presentation was more similar to that of conventional PV or PF, and it was found in 12 patients (54.5%) (Figure 1c,f). These patients mainly presented with discrete erythematous eroded or crusted patches. Cutaneous lesions were mainly distributed on the trunk in most cases, followed by its presence on limbs.

The percentage of mucosal involvement was not statistically different between IgG/IgA pemphigus patients with the two skin lesion types (*p* = 0.937). However, patients presenting with annular or arciform erythemas had a higher rate of negative results for IgG deposition in IIF studies than patients presenting with lesions more similar to those in conventional pemphigus (*p* = 0.029).

### 3.2. Histopathologic Features of Patients with IgG/IgA Pemphigus

The most common histopathologic feature of IgG/IgA pemphigus was intra-epidermal blisters (Figure 2a–c). These blisters varied in size and some were infiltrated and aggregated with neutrophils (nine cases, 40.9%) or eosinophils (seven cases, 31.8%). Four cases (18.2%) showed concurrent intra-epidermal aggregation of both neutrophils and eosinophils (Table 2). Acantholysis was suprabasal or intra-epidermal and was present in 17 cases (77.3%), with this feature being scanty or subtle in four cases (18.2%).

In the DIF studies, the deposition of IgG and IgA in the intercellular space of the epidermis was clearly identified at different intensities in all patients (Figure 2d–f). C3 deposition in the intercellular space of the epidermis was found in 18 patients (81.8%). The distribution of IgG, IgA, and C3 was in the whole layer, upper layer, or lower layer of the epidermis (Table 2). One case with fibrinogen, one case with both IgM and fibrinogen, and one case of C1q deposition in the intercellular space of the epidermis were also noted.

The percentages of histological acantholysis and intra-epidermal infiltration of neutrophils, eosinophils, or both were not significantly different between the two skin lesion types of patients with IgG/IgA pemphigus (*p* = 0.078, 0.096, 0.452, 0.184, respectively).

### 3.3. Patients with IgG/IgA Pemphigus Show Higher Epidermal Expression of IL-8 and MMP-9 Compared to Those with Conventional Pemphigus

To further characterize patients with IgG/IgA pemphigus and to differentiate them from those with conventional pemphigus, IHC staining of potential biomarkers, including C5a, CD89, IL-8, MPO, MMP-9, and IL-17, was performed with specimens from all IgG/IgA pemphigus cases (*N* = 22) and all conventional pemphigus controls (*N* = 20) (Figure 3a). The clinical severity of disease of the control group was similar (the median value of PDAI was 24, range 13–58; the median titer of IgG in IIF was 1:40) to that of the patients with IgG/IgA pemphigus. Staining intensities of IL-8 and MMP-9 in the epidermis were significantly higher while that of C5a in the epidermis was lower in patients with IgG/IgA pemphigus (*p* < 0.05). The number of infiltrating cells containing these biomarkers in the dermis was slightly higher in patients with IgG/IgA pemphigus. However, these differences were not statistically significant (Figure 3b).

### 3.4. Treatments for Patients with IgG/IgA Pemphigus

The mainstay treatment for patients with IgG/IgA pemphigus in this study was systemic corticosteroids (100%) with a starting dose of 0.5–1.0 mg/kg/day prednisolone equivalent (Table 1). Other medications may be used in combination with corticosteroids, including azathioprine (four cases, 18.2%), rituximab (four cases, 18.2%), methotrexate (one case, 4.5%), and dapsone (one case, 4.5%). The clinical course of the patients with IgG/IgA pemphigus was benign. Most responded well to the treatments and achieved complete remission within six months (16/22, 72.7%; six cases were lost to follow-up).

## 4. Discussion

Several characteristic features of patients with IgG/IgA pemphigus could be identified from our study, especially when compared to previously published case series (Table 3) [5,6,7]. In brief, IgG/IgA pemphigus tends to affect middle-aged persons irrespective of their sex. The cutaneous lesions may have two main presentations. One is annular or arciform erythemas with blisters, which may be present in 30.0–45.5% of patients. The other is discrete erosions or blisters just as those in conventional pemphigus. Mucosal involvement is common and may be observed in 40.0–61.5% of patients. In histopathology, acantholysis may not be discernible in all cases and may be identified in 17.4–100% of reported patients. Intra-epidermal infiltrations and aggregates of neutrophils or eosinophils may be found in 40.9–76.9% and 23.3–38.5% of patients, respectively. In DIF studies, almost all cases showed positive depositions of IgG and IgA in the intercellular space of the epidermis and a high rate (81.8–100%) of C3 deposition.

The annular or arciform erythemas in IgG/IgA pemphigus raise the question whether IgG/IgA pemphigus is a distinct disease entity or whether it is only a special form of pemphigus herpetiformis or IgA pemphigus. We believe that IgG/IgA pemphigus is distinct from pemphigus herpetiformis for several reasons. First, clinically, mucosal involvement has been reported but is only observed in about 8% of patients with pemphigus herpetiformis [9]. The rate is much lower than that in IgG/IgA pemphigus. Second, histopathologically, intra-epidermal infiltration in pemphigus herpetiformis is mainly by eosinophils rather than by neutrophils. A previous review [9] showed that intra-epidermal infiltration of eosinophils in pemphigus herpetiformis was present in 84% of the cases, while neutrophils could only be found in 38%. In contrast, the rate of intra-epidermal infiltration by neutrophils is higher than that by eosinophils in IgG/IgA pemphigus. Third, in DIF studies, the positive rates of deposition of IgG, IgA, and C3 in intercellular space of the epidermis are very high in IgG/IgA pemphigus. However, the positive rates of deposition of IgA and C3 in the intercellular space of the epidermis in pemphigus herpetiformis are 6.3% and 44.4%, respectively [9]. In addition, although neutrophil infiltration seems to play a role in the development of both IgG/IgA pemphigus and IgA pemphigus, we believe that IgG/IgA pemphigus is different from IgA pemphigus in several aspects. First, the rate of mucosal involvement in IgA pemphigus has been reported to be about 13.2% [10] and is much lower than that in IgG/IgA pemphigus. Second, in DIF studies, the positive rates of deposition of IgG and C3 in the intercellular space of the epidermis in IgA pemphigus are 20.5% and 13.6%, respectively [10]. Based on these observations, we propose that IgG/IgA pemphigus is a distinct disease entity rather than a subtype or a transitional form of other pemphigus types.

The results of IHC staining also support the notion that IgG/IgA pemphigus is distinct from conventional pemphigus. IgG/IgA pemphigus showed higher expression of IL-8 and MMP-9 in the epidermis than conventional pemphigus. IL-8 is a major chemoattractive factor for recruiting neutrophils [11,12]. Therefore, the intra-epidermal aggregation of neutrophils in IgG/IgA pemphigus is reasonable. Another chemoattractive factor is C5a, which has been shown to play an important role in the pathogenesis of bullous pemphigoid, especially in neutrophil recruitment [13,14]. However, its role in pemphigus is still undetermined and seldom addressed. We found that the epidermal expression of C5a in IgG/IgA pemphigus was lower than that in conventional pemphigus, which might result from the inability of IgA antibodies to activate the completement system through the classical pathway [15]. On the other hand, the Fc portion of IgA can bind to its receptor, FcαR (CD89), and subsequently activate the cells bearing these receptors [16]. CD89 can be detected on neutrophils, eosinophils, monocytes/macrophages, and some dendritic cells [16,17]. IgG/IgA pemphigus showed slightly but not significantly higher expression of CD89 compared to conventional pemphigus. This might have been due to intra-epidermal aggregates of neutrophils and eosinophils identified in 40.9% and 31.8% of the IgG/IgA pemphigus patients, respectively, and might be supported by the results of MPO staining, which also showed slightly but not significantly higher expression in IgG/IgA pemphigus.

IL-17 has been demonstrated to have a role in pemphigus [18,19]. The inhibition of the IL-23/IL-17 axis has been shown to have therapeutic potential in pemphigus [20]. IL-17 is mainly produced by T-helper 17 (Th17) cells, but it can also be produced by natural killer cells, γδ T cells, and even neutrophils [19]. We found that the expression of IL-17 was not significantly different among these patients but showed a slightly higher level in patients with IgG/IgA pemphigus. In contrast, the epidermal expression of MMP-9 was significantly higher in patients with IgG/IgA pemphigus. MMP-9 can cause degradation of the extracellular matrix and contribute to the subsequent blister formation; it can be produced by neutrophils and eosinophils [21]. Our results for MMP-9 support the pathogenic role of infiltrating neutrophils and eosinophils in the pathogenesis of IgG/IgA pemphigus.

However, IgG/IgA pemphigus shows heterogeneous clinical presentation. The reason for these patients showing different skin lesions is beyond the scope of this study and also cannot be explained by the current data. Previous studies [6,7] have shown that IgG and IgA antibodies in patients with IgG/IgA pemphigus could target Dsg1, Dsg3, and/or desmocollin 1 (Dsc1)–Dsc3 to a different extent, which may account for the mixed clinical presentations of the disease. In addition, one study [22] demonstrated that autoreactive IgG4 and IgA B cells may evolve through distinct subclass switch pathways in pemphigus, which means different autoreactivity or clonality of these IgG4 and IgA antibodies in pemphigus. Moreover, the results from a proteomic analysis of pemphigus autoantibodies revealed a larger, more diverse, and more dynamic repertoire than determined by B cell genetics [23]. These observations provide insights into the heterogeneous presentation of IgG/IgA pemphigus. Hence, based on the data from our study and previous studies, we propose that IgG/IgA pemphigus is a distinct type of pemphigus with limited but heterogeneous clinical presentation.

An association with malignancies has been reported for 9.1–27.0% of patients with IgG/IgA pemphigus (Table 3). Similarly, malignancies have been reported for approximately 11% of patients with PV [5] and 18% of patients with IgA pemphigus [10]. Whether these are true associations or just a coincidence needs further investigation. The mainstay treatment for patients with IgG/IgA pemphigus is systemic corticosteroids. Dapsone, which is frequently used in IgA pemphigus and other types of autoimmune blistering diseases due to its antineutrophilic effects [10], may also serve as a good treatment choice for patients with IgG/IgA pemphigus because it is the second most common treatment in most case series (Table 3).

This study has several limitations. First, it is a retrospective study, and the number of cases is relatively small. Second, we could not obtain the sera of the patients to further analyze the profiles of autoantibodies and their target antigens. Third, we only selected certain biomarkers, which we viewed as more relevant to the disease, to perform IHC staining rather than using an unbiased and more comprehensive method of examining the difference between IgG/IgA pemphigus and conventional pemphigus. Future studies are warranted to confirm our observations, to explore the mechanisms of heterogeneity of the disease, and to establish a consensus for its clinical presentation and histopathologic features and therapeutic approaches to it.

## 5. Conclusions

In conclusion, we propose that although IgG/IgA pemphigus shows heterogeneous clinical and histopathologic presentations, it should be classified as a distinct type of pemphigus rather than as a transitional form or a subtype of other pemphigus types.

## Figures and Tables

**Figure 1 biomedicines-10-01197-f001:**
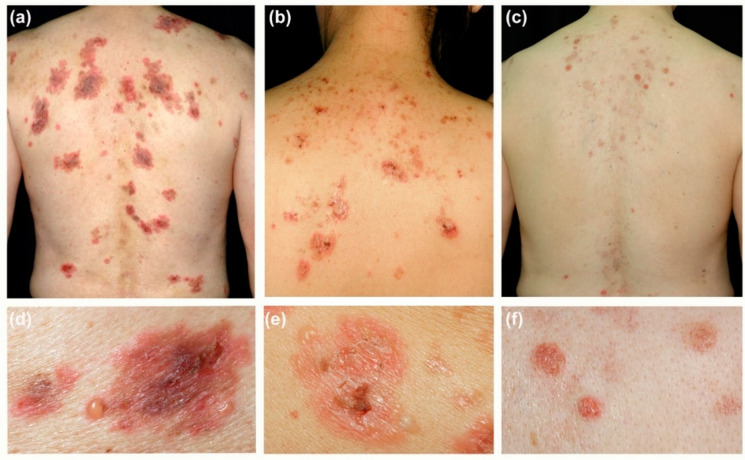
Clinical images of representative patients with IgG/IgA pemphigus. (**a**,**b**,**d**,**e**) Annular or arciform erythematous patches with vesicles, pustules, or crusted wounds found in some patients. (**c**,**f**) Other patients may present with discrete erythematous eroded macules or patches similar to those in conventional pemphigus.

**Figure 2 biomedicines-10-01197-f002:**
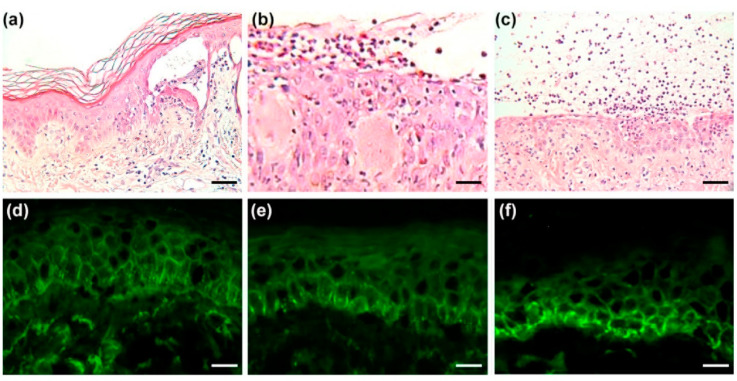
Features of histopathology and direct immunofluorescence in patients with IgG/IgA pemphigus**.** (**a**) Intra-epidermal blisters with scanty acantholysis and apoptotic keratinocytes were found (Scale bar = 50 μm). (**b**) Infiltration of eosinophils and dermal edema was noted in some cases (Scale bar = 25 μm). (**c**) Infiltration of neutrophils was sometimes prominent (Scale bar = 50 μm). (**d**) Positive IgG deposition in the intercellular space of the epidermis was shown in direct immunofluorescence (Scale bar = 50 μm). (**e**,**f**) Positive IgA depositions at lower levels of the epidermal intercellular space were demonstrated by direct immunofluorescence (Scale bar = 50 μm).

**Figure 3 biomedicines-10-01197-f003:**
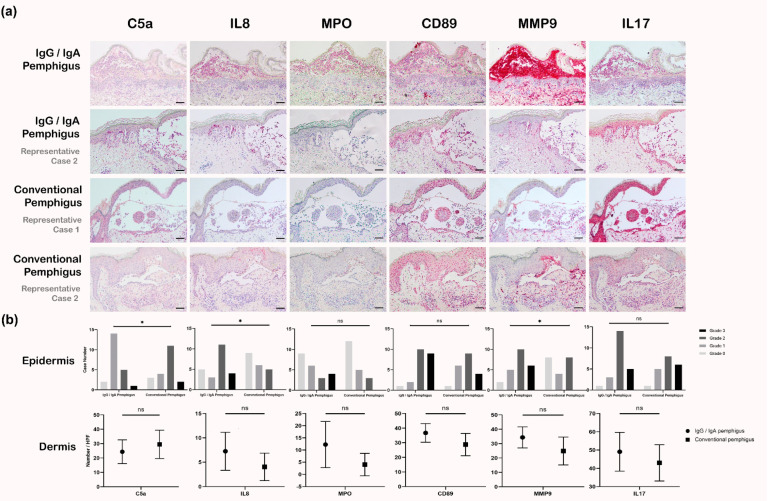
Comparisons of immunohistochemical stains between patients with IgG/IgA pemphigus and those with conventional pemphigus. (**a**) Representative images of stains from two cases of IgG/IgA pemphigus and two cases of conventional pemphigus (Scale bar = 100 μm). (**b**) Patients with IgG/IgA pemphigus (*N* = 22) show significantly higher expression of IL-8 and MMP-9 and lower expression of C5a in the epidermis compared to patients with conventional pemphigus (*N* = 20) (*, *p* < 0.05, ns, non-significance).

**Table 1 biomedicines-10-01197-t001:** Demographics and clinical presentation of patients with IgG/IgA pemphigus.

	Age	Sex	Clinical Lesions		IgG IIF	PDAI	Comorbidity	Treatment
			Skin ^#^	Mucosa				
1	44	F	2	+	1:40+	34		S
2	72	M	1	−	−	16		S
3	23	F	2	−	1:20+	23		S
4	69	F	2	−	1:20+	21	HTN, CAD	S
5	56	F	2	−	1:10+	17	CU	S
6	37	M	2	+	1:20+	27		S
7	63	F	1	+	1:40+	18	CU, hyperlipidemia	S
8	39	F	1	−	1:20+	16		S, D
9	49	M	2	−	1:40+	12		S
10	48	M	1	−	1:160+	20	Esophageal cancer	S
11	47	M	2	+	1:80+	21		S, A
12	56	F	1	−	1:20+	15	HTN	S
13	47	F	1	+	−	28	DM	S, R, A
14	37	M	2	+	1:20+	19		S, A
15	68	M	2	+	1:160+	47	HTN	S, R
16	28	F	1	−	−	9		S
17	82	M	2	−	−	11		S
18	67	M	2	−	1:20+	14	CAD	S
19	37	F	2	−	1:20+	6		S
20	69	F	1	+	−	29	HTN	S, R, A
21	41	F	1	−	1:20+	19	SLE, asthma	S, MTX
22	67	M	1	+	−	38	Colon cancer	S, R

A: azathioprine; CAD: coronary artery disease; CU: chronic urticaria; D: dapsone; DM: diabetes mellitus; F: female; HTN: hypertension; IIF: indirect immunofluorescence; IgG: immunoglobulin G; M: male; MTX: methotrexate; PDAI: pemphigus disease area index; R: rituximab; S: steroids; SLE: systemic lupus erythematosus. # Two types of skin lesions: 1. annular or arciform erythemas with blisters or erosions; 2. blisters or erosions as seen in conventional pemphigus.

**Table 2 biomedicines-10-01197-t002:** Results of histopathology and direct immunofluorescence.

	Histopathology			DIF at Intercellular Space *						
	Acantholysis ^#^	Intra-Epidermal Aggregates		IgG		IgA		C3		
		Neutrophils	Eosinophils							
1	+	−	+	++	W	++	W	++	W	
2	−	+	+	++	W	++	W	+	W	
3	+	+	−	++	W	+	W	−	−	
4	−	−	+	++	W	++	W	++	W	
5	+	−	−	++	U	++	U	+	U	
6	+	−	−	++	W	++	U	++	W	
7	−	+	−	++	W	++	W	++	W	
8	+/−	+	+	++	L	+	L	++	L	
9	+/−	−	−	++	W	+	W	++	L	
10	+/−	−	+	++	L	+	L	++	L	
11	+/−	−	−	++	W	++	W	+	L	
12	−	+	+	++	W	+	W	++	L	
13	+	−	−	++	L	++	L	+	L	
14	+	+	−	++	W	++	W	−	−	
15	+	−	−	++	L	++	L	++	L	
16	−	−	−	+	U	+	U	+	U	
17	+	−	−	++	L	+	L	++	L	
18	+	−	−	++	W	+	L	++	L	
19	+	+	+	++	W	++	W	+	L	
20	+	−	−	++	W	++	W	−	−	
21	+	+	−	++	W	++	W	−	−	
22	+	+	−	++	W	++	W	++	W	

DIF: direct immunofluorescence; IgA: immunoglobulin A; IgG: immunoglobulin G; L: lower layer; U: upper layer; W: whole layer. # The symbol +/- (trace) means only scanty acantholysis was found in the specimens. * The intensity of DIF staining is indicated as follows: +, weaker staining; ++ regular staining. The distribution of staining is indicated as follows: W, whole layer of epidermis; U, upper portion of epidermis; L, lower portion of epidermis.

**Table 3 biomedicines-10-01197-t003:** Comparison of case series of IgG/IgA pemphigus.

	Current Study *N* = 22	Toosi S, et al., 2016 [5] *N* = 13	Hashimoto T, et al., 2018 [6] *N* = 30	Criscito MC, et al., 2021 [7] *N* = 44
**Age (years)**	52.1	48.5	55.6	59.0
**Sex (M/F)**	10/12	7/6	15/13	21/20
**Clinical presentation**				
Annular erythemas	45.5%	N/A	30.0%	43.9%
Mucosal lesions	40.9%	61.5%	46.7%	40.0%
**Histopathology**				
Acantholysis	77.3%	84.6%	17.4%	100%
Intra-epidermal infiltration				
Neutrophils	40.9%	76.9%	47.8%	44.2%
Eosinophils	31.8%	38.5%	34.8%	23.3%
**DIF at intercellular space**				
IgG	100%	100%	77.3%	100%
IgA	100%	100%	77.3%	97.5%
C3	81.8%	100%	83.3%	N/A
**Comorbidities**				
Malignancy	9.1%	15.4%	20.0%	27.0%
Autoimmune	9.1%	7.7%	6.7%	10.8%
**Treatment**				
Systemic steroids ^#^	95.5%	N/A	52.6%	59.0%
Dapsone	0.0%	15.4%	26.3%	15.4%
Steroids + Dapsone	4.5%	15.4%	5.3%	17.9%

DIF: direct immunofluorescence; N/A: not applicable; # systemic steroids alone or in combination with immunomodulators other than dapsone.

## Data Availability

The data presented in this study are available upon request from the corresponding author. The data are not publicly available due to privacy issues.

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
