# Peer review of "Clinical, Histopathologic, and Immunohistochemical Features of Patients with IgG/IgA Pemphigus"

_biomedicines, 2022, doi:10.3390/biomedicines10051197_

Round 1

Reviewer 1 Report

The authors reported a case series of IgG/IgA pemphigus, which were mainly diagnosed based on direct/indirect immunofluorescence.

The novelty of this manuscript seems weak. And, the evidence of IgG/IgA pemphigus seems weak. Are there any relations between the clinical manifestations of arcuate erythema with erosions and the positivity of IgA-IIF? How was the result of IgA-IIF in these cases? How was the result of CLEIA/ELISA for anti-desmoglein 1/3 IgG? It is favorable to show western blotting results to prove circulating IgG/IgA reacting to desmogleins/desmocolins.

One of the novelty of this case series is that IL-8 and MMP-9 staining intensities are higher in the epidermis. First of all, the evaluation of staining intensity generally lacks objectivity, and the authors should show representative images of each grade in the manuscript. Who performed the staining, and who evaluated the intensity? Are the images blinded and randomized, and did multiple evaluators score the intensity?  Also, the details of the methods of staining is lacked.

Nevertheless these points are solved, it is still doubtful to prove the novelty; the elevation of IL-8 and MMP-9 seems to be just a result of neutrophil accumulation as a general phenomenon.

Reviewer 2 Report

It was a pleasure reading this paper on an uncommon autoimmune bullous disorder of the pemphigus group, IgG / IgA pemphigus. The authors retrospectively collected data from 22 patients with this diagnosis confirming the findings from previously published case series. Their immunohistochemical study is a novel and interesting addition to this topic.

The article is overall well written but there are several spelling mistakes which the authors will be able to correct during revision. The methodology is sound and clearly explained and the results, that are presented together with high quality clinical and microscopic images and tables, allow the authors to support that IgG / IgA pemphigus is a separate disease entity rather than a subtype of other forms of pemphigus.

Please consider my following comments to further improve this manuscript:

  • Page 3 line 98: “the average age”: I suggest reporting the median age and range.
  • Page 5 line 144 “at different levels”: please describe at which is the level of IgA deposition in Figures 2d, e, and f, according to the terminology used in Table 2.
  • Page 6 line 167 “were performed”: please clarify whether the immunohistochemistry study was performed on samples from all the IgG/IgA pemphigus cases and all the conventional pemphigus controls or just on a subgroup of subjects from the two groups.
  • Page 9 line 285 “Dapsone …”: This is usually the drug of choice in IgA pemphigus. I suggest adding a brief statement on the rationale for its use, citing the relevant literature.

Reviewer 3 Report

The authors Cho et al present a single-center analysis of the rare autoimmune bullous condition of IgG/IgA pemphigus which has been described previously as an overlap between pemphigus vulgaris and IgA pemphigus. This condition is very rare, and therefore a case series of 22 patients is of significant value, this number compares to previously published articles. The article is well structured and the results support the conclusions drawn. The topic is also of interest to the readership as autoimmune bullous diseases serve as models for drug development in dermatology. It will be important in the future to study rare subtypes like IgG/IgA pemphigus in more detail.

The methods used in this case series are quite basic, yet, the analysis of potential biomarkers adds further value to the article. The figures and tables are of good quality and add value to the article.

Overall I recommend publication but a few issues need to be addressed as follows:

Major revisions:

- Revision for English language / grammar throughout

Minor revisions:

- it appears that the tables are not exactly in MDPI style (font?)

- table 1: Rx is not explained as abbreviation

Round 2

Reviewer 1 Report

Although the revewer does not completely agree with the accuracy of diagnosis and the controls are still lacked, the revised manuscript is improved. And the authors responded to the reviewers' comments.